# Structural, Optical, Electrical, and Thermoelectric Properties of Bi_2_Se_3_ Films Deposited at a High Se/Bi Flow Rate

**DOI:** 10.3390/nano13202785

**Published:** 2023-10-18

**Authors:** Ya-Hui Chuai, Yun-Fan Wang, Yu Bai

**Affiliations:** School of Physics, Changchun University of Science and Technology, 7089 Satellite Road, Changchun 130022, China; wyf1033539583@163.com (Y.-F.W.); baiyu@cust.edu.cn (Y.B.)

**Keywords:** Bi_2_Se_3_ film, semiconductor thermoelectric materials, PECVD, transmittance, thermoelectric properties

## Abstract

Low-temperature synthesis of Bi_2_Se_3_ thin film semiconductor thermoelectric materials is prepared by the plasma-enhanced chemical vapor deposition method. The Bi_2_Se_3_ film demonstrated excellent crystallinity due to the Se-rich environment. Experimental results show that the prepared Bi_2_Se_3_ film exhibited 90% higher transparency in the mid-IR region, demonstrating its potential as a functional material in the atmospheric window. Excellent mobility of 2094 cm^2^/V·s at room temperature is attributed to the n-type conductive properties of the film. Thermoelectrical properties indicate that with the increase in Se vapor, a slight decrease in conductivity of the film is observed at room temperature with an obvious increase in the Seebeck coefficient. In addition, Bi_2_Se_3_ thin film showed an enhanced power factor of as high as 3.41 μW/cmK^2^. Therefore, plasma-enhanced chemical vapor deposition (PECVD)-grown Bi_2_Se_3_ films on Al_2_O_3_ (001) substrates demonstrated promising thermoelectric properties.

## 1. Introduction

Thermoelectric materials-based technologies are attracting the attention of researchers due to their direct heat to electricity conversion capabilities [1,2,3,4]. These modern and green technologies can fulfill the ever-growing energy demands, one of the biggest challenges of the modern world, via efficient heat-capturing capacity and energy conversion. The efficacy of these materials is measured using a dimensionless figure of merit ZT (S^2^σT/κ), where S stands for the Seebeck coefficient for electrical conductivity, T for absolute temperature, and κ for thermal conductivity [5]. The ZT of materials has been improved through the development of multiple techniques. Among these methods, maximizing the performance factor PF (S^2^σ) has been partly successful. PF is improved through quantum confinement effects or the reduction of thermal conductivity through phonon scattering at grain boundaries [6,7].

Bi_2_Se_3_ is a thermoelectric semiconductor frequently used in applications requiring low-temperature power production (<600 K), refrigeration, and power generation. Since low- and mid-temperature ranges of the Earth’s environment are the most abundant waste heat source [8,9,10], Bi_2_Se_3_ is employed as an effective and affordable form of exploration to recover low-grade waste heat. Due to its naturally small PF and extremely low ZT, it is extensively used in Internet of Things-based devices and self-powered electronics [11,12].

Several previously reported studies have shown that PF can be enhanced by broadening the bandgap ($E_{g}$) and correspondingly decreasing the σ [13,14]. Another approach to improve the material performance is by optimizing its charge carrier concentration [15,16]. Two main strategies are employed to achieve this: tuning the stoichiometry of materials and introducing atomic impurities. Doping of Bi_2_Se_3_ with different elements, such as Sn, Ni, Te, and Sb, has been reported to improve its thermoelectric performance [17,18,19,20,21,22,23]. However, the application is far from practical. Moreover, the effect of an optimized stoichiometric ratio on improving the thermoelectric performance of Bi_2_Se_3_ has not been explored in detail.

A high-yield and low-temperature synthesis method for preparing high-quality Bi_2_Se_3_ films is proposed herein. The photoelectric and thermoelectric properties, structure, and morphology of the prepared films in a rich Se environment are studied. In addition, the effect of Se vapor on film growth is evaluated, and results revealed that higher pressure yielded high-performance Bi_2_Se_3_-based materials. Thus, the as-synthesized Bi_2_Se_3_ films demonstrated high application potential in environmentally friendly large-area thermoelectric devices.

## 2. Experimental

### 2.1. Preparation of Bi_2_Se_3_ Thin Films

A well-known plasma-enhanced chemical vapor deposition (PECVD) method was used to prepare Bi_2_Se_3_ thin films on Al_2_O_3_ (001) substrates. Trimethyl bismuth (TMBi, 98%) and selenium diethyl ester (DESe, 98%) were used as Bi and Se sources. Hydrogen (H_2_), as a carrier gas, was passed through the TMBi and DESe liquid in a bubbler. The prepared thin films’ physical properties were studied to compare two samples. Sample F1: the flow rate of H_2_ carrier gas was 20 sccm and 60 sccm, and Bi and Se vapor flow rates were 1:3 (the TMBi and DESe flow rates were 0.2 sccm and 0.6 sccm, respectively), at a total pressure of 40 Pa. Sample F2: the flow rates of H_2_ carrier gas were 20 sccm and 100 sccm, and the Bi and Se vapor flow rates were 1:5 (the TMBi and DESe flow rates were 0.2 sccm and 1 sccm, respectively), at a total pressure of 65 Pa. During film growth, the pulse generator DC power was adjusted to 200 W with a pulsing frequency of 20 kHz, maintaining the stability of the plasma source. The desired film thickness of 200 nm was achieved after 40 min of reaction, followed by annealing in high purity N_2_ at 150 °C for 30 min to remove H impurities in the films. The finished films were then stored for later use and progressively cooled to room temperature.

### 2.2. Characterization of Prepared Thin Films

#### 2.2.1. Structural Characterization

The crystallinity of the as-synthesized Bi_2_Se_3_ films was studied by X-ray diffraction (XRD) diffractometer (Bruker D8 Advance XPert), with a Cu-Ka and λ 1.540 Å. Using a 514.5 nm line Ar+ laser, room temperature Raman spectra were acquired. The valence states of the elements were examined using X-ray photoelectron spectroscopy (XPS, ESCALAB 250). The film surface morphology and atomic organization were investigated using an atomic force microscope (AFM Veeco DI-3100). A high-resolution transmission electron microscope (HRTEM, TEM 2010F) was used to observe the inner morphology and thickness of films. 

#### 2.2.2. Electrical and Thermoelectric Characterization

A Hall-effect device (ACCENT HL55OOPC) was used to investigate the carrier concentration (*n*) and mobility (μ) of the prepared films in the temperature range of 300 to 700 K. Thermoelectric properties were studied by a conductivity/Seebeck coefficient instrument (NETZSCH Sba-458). Different thermoelectric parameters, such as power factor (PF), Seebeck coefficient (S), and electric conductivity (σ), were determined. Ar was used to maintain a positive pressure gas environment in the chamber to prevent the oxidation of the film during the temperature measurement. Four probes and a thermocouple were set up in the equipment, and a four-wire method was used to measure the conductivity of the films. When one end of the sample was heated with a heating wire, the temperature difference (ΔT) between the two ends was produced. Thus, the voltage difference (ΔV) was measured. Finally, the Seebeck coefficient of the thin films was determined by fitting the ΔU-ΔV curve.

#### 2.2.3. Optical Characterization

FTIR spectrometer (Thermo Nicolet iS50) was employed to study the transmission and absorption of the film. Absorption spectra are measured in the wavelength range of 2–6 μm. The transmission spectrum is measured in the mid-infrared wavelength range of 3–5 μm.

## 3. Results and Discussion

### 3.1. XRD and Raman Analysis

The crystallinity of Bi_2_Se_3_ films on Al_2_O_3_ (001) substrates was determined by XRD. Figure 1a,b shows the XRD patterns of Bi_2_Se_3_ films at different Bi and Se pressure ratios. High-intensity peaks (006), (009), (0012), (0015), (0018), and (0021) indicate the good crystallinity of films. Diffraction peaks (001) reveal the preferential orientation of films, perpendicular to the substrate surface along the *c*-axis, indexed as rhombohedral Bi_2_Se_3_ phase (JCPDS card no.: 330214) [24]. The circle depicts the peak position of Bi in Figure 1a, suggesting the excess of Bi in sample F1, grown under the condition of Bi and Se vapor ratio of 1:3. Figure 1a,b shows the full width at half-maximum (FWHM) of the (009) rocking curve to be approximately 1.2° and 0.28°, respectively, showing that the F2 film has higher crystallinity due to its higher Bi and Se vapor ratio of 1:5. The lattice parameters *a*, *b*, and *c* for the Bi_2_Se_3_ films are derived from the d-spacings of the (009) peak shown in Table 1. These patterns match those in the normal card data file (No. 01-089-2008).

A 514.5 nm-wavelength laser was used for the structural quality of the as-synthesized thin films at room temperature. According to the group theory, the Bi_2_Se_3_ thin films exhibit two modes; active infrared mode (ungerade) and Raman active mode (gerade), as represented by the following equation; Γ = 2E_g_ + 2A_1g_ + 2E_u_ + 2A_1u_, where g and u indicate gerade and ungerade modes, respectively. $A_{1g}^{1}$ (out-of-plane), $A_{1g}^{2}$ (out-of-plane), and $E_{g}^{2}$ (in-plane) are the three Raman active modes for Bi_2_Se_3_ films [25]. Based on the relative motion of Bi and Se atoms divided by a van der Waals gap, five atomic layers (Se1-Bi-Se2-Bi-Se1) were obtained (Figure 2a) [26]. The Raman spectra of samples F1 and F2 are shown in Figure 2b,c, respectively. For both samples, three distinct peaks labeled $A_{1g}^{1}$, $A_{1g}^{2}$, and $E_{g}^{2}$ modes are observed in low-wave numbers. For sample F1, the FWHM values for active Raman modes $A_{1g}^{1}$, $E_{g}^{2}$, and $A_{1g}^{2}$ modes are 73.5 (12.7), 132.6 (4.9), and 175.8 (8.5) cm^−1^, respectively. For sample F2, the FWHM values are 73.2 (4.3), 132.6 (4.1), and 175.8 (6.6) cm^−1^, respectively. These peak positions are consistent with previously reported bulk and single-crystal Bi_2_Se_3_ compounds [27]. Smaller FWHM indicates higher crystallinity of sample F2 than sample F1, consistent with the previous XRD conclusion. This shows that the film prepared under high Se vapor has excellent crystallinity. The Scherrer equation calculated the average crystal sizes of Bi_2_Se_3_ films:τ = kλ/βcos*θ*(1)
where k indicates the shape factor, β is FWHM, λ shows the wavelength of incident X-rays, and θ represents the diffraction angle (corresponding to the (009) plane) [28]. For samples F1 and F2, the average grain sizes of the films are 100 and 121 nm, respectively.

### 3.2. XPS Analysis

XPS investigates the valence state and elemental composition of Bi_2_Se_3_ films. The XPS spectra of samples F1 (a1) and (b1) and samples F2 (a2) and (b2) are shown in Figure 3, indicating the existence of Bi and Se in the films. Characteristic spin-orbit coupling value of 5.3 eV along with XPS peaks corresponding to Bi-4f_7/2_ (158.7 eV) and Bi-4f_5/2_ (164 eV), indicate the presence of Bi^+3^, as shown in Figure 3a1,a2 [28]. Bi-4f_7/2_ and Bi-4f_5/2_ peaks around 160.3 eV and 165.4 eV, respectively, are attributed to Bi (Figure 3a1), confirming excessive Bi in sample F1 consistent with the previous XRD test results. For the core level Se-3d, narrow scan, a broad peak with a shoulder appeared as the 3d_5/2_ and 3d_3/2_ signals, owing to the narrow spin-orbit coupling value of ~0.7 eV, for samples F1 and F2, respectively (Figure 3b1,b2). The deconvolution XPS spectra confirmed the respective 3d_5/2_ and 3d_3/2_ peaks at 53.6 eV and 54.3 eV, suggesting the presence of Se^−2^ [29].

Table 2 shows the atomic concentrations of Bi_2_Se_3_ films, confirming the presence of excessive Bi (or lack of Se) in sample F1.

The atomic concentration ratio and binding energies of Se and Bi in sample F2 are comparable to those found in bulk pure Bi_2_Se_3_ single crystal. These results suggest that high-purity Bi_2_Se_3_ thin films are successfully deposited on Al_2_O_3_ (001) substrate via the PECVD method.

### 3.3. AFM and HRTEM Analysis

The surface morphology of F1 and F2 samples is shown in Figure 4a1,a2, respectively. These pictures show the granular morphology of all the deposited thin films, with small triangular granules evenly dispersed throughout the film surfaces. For both samples, the root mean square (RMS) values for surface roughness are 7.7 and 4.9 nm, respectively. Surface roughness decreased and grain size increased with Se vapor, suggesting a decrease in grain boundaries. 

This shows the better crystallization quality of the film prepared at high Se pressure. The development of thin films from deposited atoms is regulated by an intrinsically non-equilibrium process, which pits thermodynamics against reaction kinetics. Two potential modes in the growth kinetics of Bi_2_Se_3_ grains are ripening and coalescence. Ripening refers to the mass transport from smaller to larger grains via surface diffusion, thereby reducing the surface area. In contrast, coalescence is the formation of larger size grains by the combination of smaller-sized grains. The delicate interaction between atomic position and surface electronic structure reconstructs a complicated surface, rearranging the atoms in the top layer to reduce surface energy.

The spacing between adjacent planes for both samples is revealed by HRTEM and selected area electron diffraction (SAED) images to be 0.2 nm, which fully conforms with the pdf file No. 01-089-2008, which correlates to the d-spacing of [11$\bar{2}$0]. The layered form along the *c*-axis is consistent with the selected area electron diffraction (SAED) pattern of the Bi_2_Se_3_ film, which shows three different diffraction spacings ([11$\bar{2}0$], [$\bar{1}2\bar{1}0$], and [$\bar{2}$110]) that are indexed as the 6-fold symmetry [001] zone axis pattern [30].

### 3.4. Optical Characteristics of Films

Figure 5a,b show the absorption spectra of Bi_2_Se_3_ films. Significant absorption edges are observed at about 2000 nm, demonstrating the fundamental absorption attributed to electronic excitations from the valence band to the conduction band. The relationship between the optical absorption coefficient (*α*) and photon energy (*hv*) is expressed by the following equation:(2)$$\alpha h\nu =A{\left(h\nu -E_{g}\right)}^{m}$$
where *A* is a constant, $E_{g}$ denotes the optical bandgap of the material, and m varies with the electronic transition, i.e., *m* = 1/2 for the direct band and *m* = 2 for an indirect band [31].

The (αhν)^2^-*hν* plot is illustrated in Figure 5a,b. The linear correlation between (αhν)^2^ and *hν* reveals the direct energy bandgap of Bi_2_Se_3_ films.

The straight region of the curve predicts that the direct optical bandgap is 0.35 eV in the vicinity of 0.33–0.4 eV. Sample F1 exhibits an average transmittance of greater than 85% in the wavelength range of 3–5 µm. Sample F2 shows an average transmittance is more than 90% in the same wavelength range. The high absorption (or low transparency) is associated with carrier transition between the conduction and valence bands in the short wavelength region. The higher transmittance of sample F2 than that of sample F1 might be attributed to high free carrier absorption and larger diffuse reflection of incident light on the surface of sample F1. We obtained a thin film material with high transmittance at the atmospheric window (3–5 μm).

### 3.5. Electrical Characteristics

Different temperature Hall effect measurements were conducted in the high-temperature range to study the transmission and electrical characteristics of as-synthesized films. The change in carrier mobility and density with temperature were investigated. For samples F1 and F2, the Hall coefficients were −13.1 and −7.2 cm^3^ C^−1^ at room temperature, respectively, depicting characteristic n-type conduction [32]. Figure 6a displays the temperature-dependent carrier concentration, which demonstrates a rise in electron concentration with temperature. This is attributed to high intrinsic excitation and impurity ionization with rising temperature. For samples F1 and F2, the electron concentration is 3.7 × 10^18^ and 6.23 × 10^17^ cm^−3^ at room temperature, respectively. The Hall mobility diminishes as temperature increases, as shown in Figure 6b. The diminished mobility is owing to the amplification of lattice vibration scattering at high temperatures. Results show that excellent Hall mobility is observed at room temperature (934 and 2094 cm^2^/V·s for two samples), nearly two orders of magnitude than the previously reported delafossite p-type infrared transparent conductive film [33,34,35]. Sample F2 has greater mobility than F1 due to its excellent crystallinity and fewer grain boundaries, which can be ascribed to the three-dimensional topological insulator structure of Bi_2_Se_3_.

Additionally, a Dirac Cone is formed by four time-reversed symmetry points in the Brillouin zone of the film surface state with Kramers’ degenerate phenomenon [36]. The Dirac point is the apex of the Dirac cone, and a linear dispersion relationship is observed between the momentum near the apex and energy. Due to the spin-coupling effect, the spin orientation of the surface state is always perpendicular to the direction of momentum; as a result, the speed of the electrons on the surface is comparable to that of the photon.

Sample F2 exhibits a higher mobility and lower carrier concentration at room temperature than F1, attributed to the high volatility of selenium (Se). Therefore, preparing Bi_2_Se_3_ film with a stoichiometric ratio is often difficult. As a result, Bi_2_Se_3_ generates Se vacancies that function as donors, producing high carrier concentration and poor carrier mobility. Significant Se atom deletion stops and pure Bi_2_Se_3_ films are formed only in the Se-rich environment.

### 3.6. Thermoelectric Characteristics

Then, thermoelectric parameters of Bi_2_Se_3_ films in the high-temperature region of 300–700 K were investigated to understand the effect of the rich Se on the electrical transport properties of thin films. According to the observations, both samples have negative Seebeck coefficients, as shown in Figure 7a, which indicates n-type conduction. For both samples, the absolute value of the Seebeck coefficient (s) improved as temperature increased. The Seebeck coefficients of samples were −128 and −174 μV/K for F1 and F2, respectively, at room temperature. Maximum values of −161 and −205 μV/K were obtained at 700 K for samples F1 and F2, respectively. Sample F2 exhibits a stronger Seebeck coefficient than F1 due to the weaker grain boundary scattering and lower carrier concentration [37].

We also investigated the influence of the compositional variations between the two samples on their thermoelectric characteristics. The deposited Bi_2_Se_3_ thin films have Seebeck coefficient values ranging from −130 to −159.2 μV/K for Se/Bi ratios between 3 and 15.45. At temperatures above 500 K, the Seebeck coefficient of the films is extremely high and stable due to the various granular shapes and compositions. Its sizes result from the thin film’s processing route. Figure 7b illustrates the electrical conductivity of Bi_2_Se_3_ films. Both samples’ electrical conductivity increases with temperature, indicating that Bi_2_Se_3_ films are semiconducting [16]. However, sample F1 has a higher conductivity than sample F2 due to fewer Se vacancies in the films prepared in the Se-rich environment. Therefore, the electron concentration of F2 is lower than that of F1. Due to the higher crystallinity, the mobility of F2 film is higher than F1. In comparison, the increase in mobility is not as fast as the decrease in electron concentration. Overall, the conductivity of sample F1 is, therefore, greater than that of sample F2. The electrical conductivities of samples F1 and F2 are 5360.8 and 4406.9 S/cm at 300 K, respectively. In semiconductor materials, there is a simple linear relationship between carrier concentration and mobility. At constant temperature, mobility decreases when the carrier concentration increases. This is due to increased scattering between carriers, which hinders their free movement through the crystal. This scattering includes the scattering between carriers and the interaction between carriers and impurity defects. In general, the higher the crystallinity of the material, the higher the carrier mobility, the smaller the carrier concentration. The conductivity of semiconductor is defined as:(3)$$\sigma =nq\mu_{n}+pq\mu_{p}$$

*n* is the electron concentration, *p* is the hole concentration, *q* is the amount of electricity, $\mu_{n}$ is electron mobility, and $\mu_{p}$ is the hole mobility. Therefore, we can conclude that the higher the carrier concentration, the greater the mobility and the higher the conductivity of the sample.

The temperature-dependent power factor (S^2^σ) for samples F1 and F2, which represents the electrical performance of thermoelectric materials, is calculated by using electrical conductivity and Seebeck coefficient. As shown in Figure 7c, owing to the reduced electrical conductivity and the enhanced Seebeck coefficient caused by Se doping, a remarkable improvement of power factor is obtained for sample F2. The maximal PF value is 3.47 μW cm^−1^ K^−2^ at 480 K, which is about 1.3 times that of sample F1. This result confirms that higher Se vapor during film preparation can enlarge the thermoelectric properties of Bi_2_Se_3_.

## 4. Conclusions

In summary, Bi_2_Se_3_ films with the preferred orientation of the *c*-axis are grown on Al_2_O_3_ (001) substrates via the PECVD method in a rich Se vapor environment. The structural and morphology analysis revealed the high quality of the stoichiometric Bi_2_Se_3_ films. The Bi_2_Se_3_ films showed significant optical transparency in the mid-IR region and superior n-type conduction. In addition, Bi_2_Se_3_ films revealed promising thermoelectric properties with a high Seebeck coefficient of −174 μV/K at room temperature and a maximum Seebeck coefficient of ~−205 μV/K at 700 K for sample F2. A maximum power factor value of ~3.47 μW/cmK^2^ was observed at 480 K for sample F2. Thus, Bi_2_Se_3_ films under higher Se vapor demonstrate beneficial thermoelectric properties with diverse application potential in optoelectric devices. The enhanced thermoelectric properties of Bi_2_Se_3_ films achieve the construction of environmentally friendly large-area thermoelectric devices.

## Figures and Tables

**Figure 1 nanomaterials-13-02785-f001:**
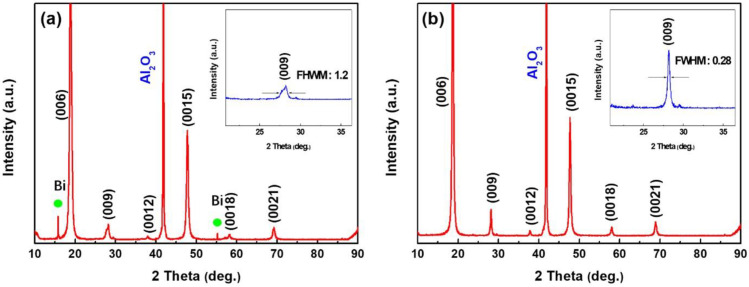
XRD pattern of Bi_2_Se_3_ films (**a**) sample F1, (**b**) sample F2, and the illustrations are enlarged XRD pattern showing (009) peak of Bi_2_Se_3_ films. (
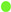
: Bi).

**Figure 2 nanomaterials-13-02785-f002:**
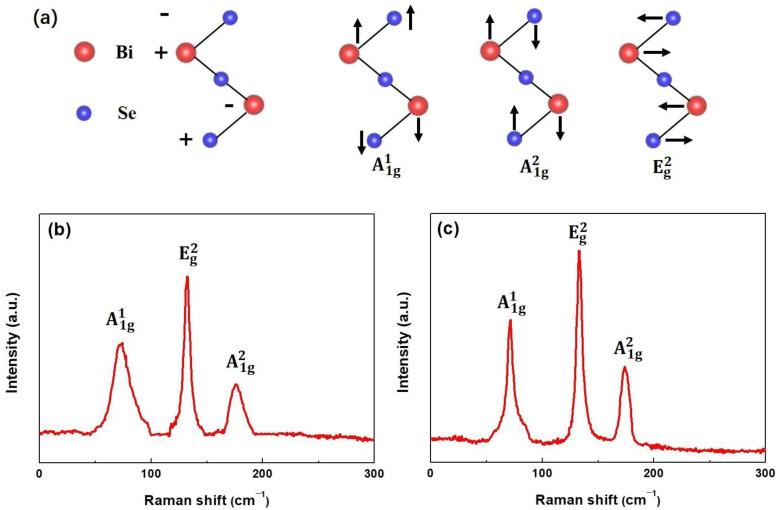
(**a**) Raman active modes: out-of-plane ($A_{1g}^{1}$ and $A_{1g}^{2}$) and i-plane $E_{g}^{2}$ for Bi_2_Se_3_ films. Room temperature Raman spectra of Bi_2_Se_3_ films (**b**) sample F1 and (**c**) sample F2.

**Figure 3 nanomaterials-13-02785-f003:**
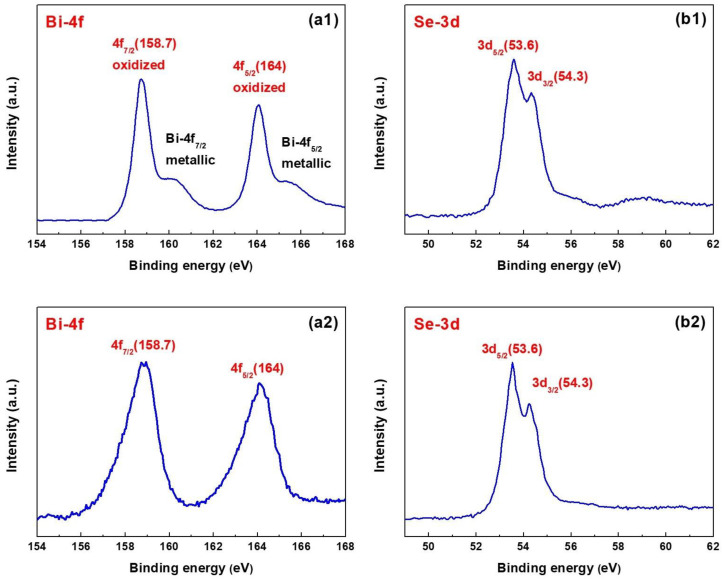
XPS of Bi_2_Se_3_ films, (**a1**) Bi-4f and (**b1**) Se-3d for sample F1, (**a2**) Bi-4f and (**b2**) Se-3d for sample F2.

**Figure 4 nanomaterials-13-02785-f004:**
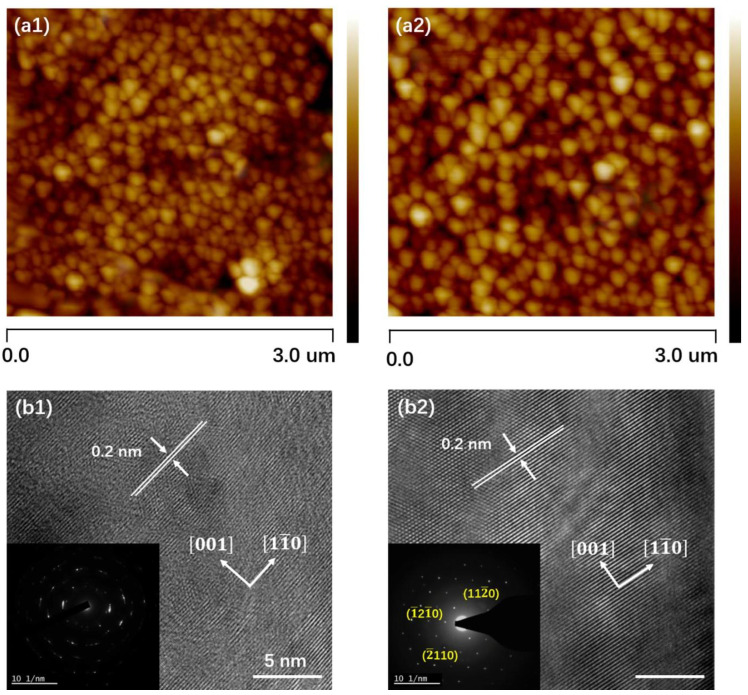
AFM, HRTEM, and SAED analyses of Bi_2_Se_3_ films, (**a1**,**b1**) for sample F1, (**a2**,**b2**) for sample F2.

**Figure 5 nanomaterials-13-02785-f005:**
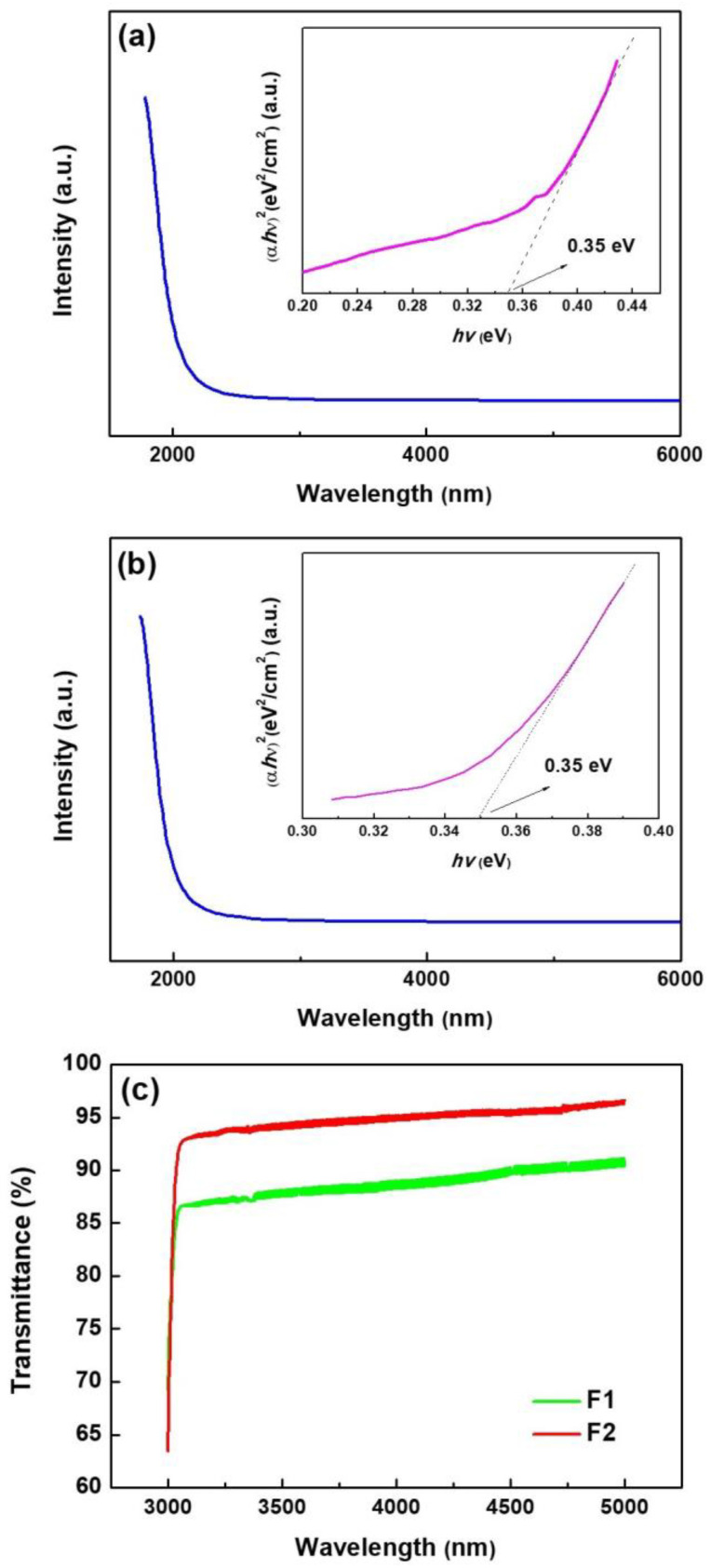
(**a**) Absorption spectrum of sample F1 and (**b**) F2, illustrations are (*ahv*)^2^ versus *hv* plot; (**c**) Optical transmission spectrum spectra of sample F1 and F2.

**Figure 6 nanomaterials-13-02785-f006:**
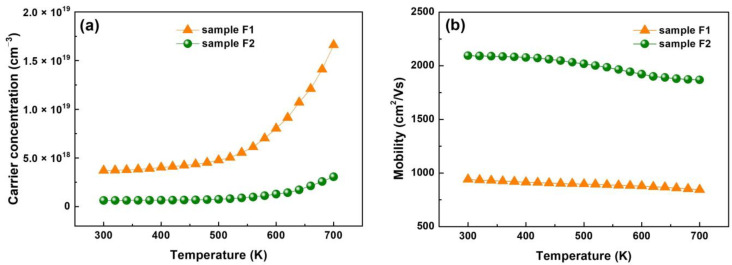
(**a**) Carrier concentration and (**b**) Hall mobility of the Bi_2_Se_3_ films prepared at different Bi and Se vapor ratios.

**Figure 7 nanomaterials-13-02785-f007:**
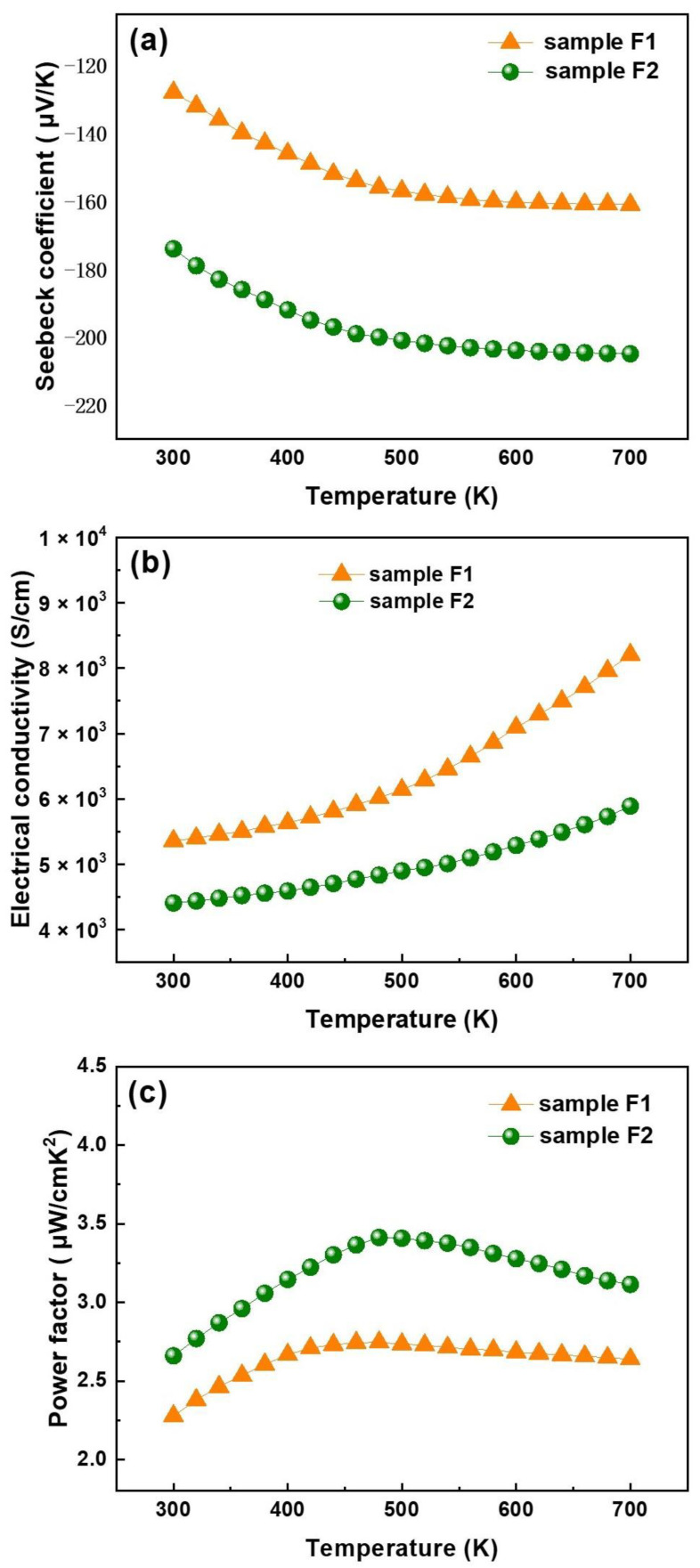
Variation in thermoelectric properties with the temperature of Bi_2_Se_3_ films: (**a**) Seebeck coefficient, (**b**) electrical conductivity, and (**c**) power factor for samples F1 and F2.

**Table 1 nanomaterials-13-02785-t001:** Lattice parameters of Bi_2_Se_3_ films prepared at different Bi and Se vapor ratios.

Sample	*a* (Å)	*b* (Å)	*c* (Å)
F1	4.121 (3)	4.121 (3)	28.605 (2)
F2	4.138 (2)	4.138 (2)	28.669 (3)

**Table 2 nanomaterials-13-02785-t002:** The atomic concentrations of Bi_2_Se_3_ films prepared at different Bi and Se vapor ratios.

Sample	Bi	Se
F1	48.91	51.09
F2	40.77	59.23

## Data Availability

All data can be made available to interest readers upon request.

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
