# Peer review of "Structural, Optical, Electrical, and Thermoelectric Properties of Bi_2_Se_3_ Films Deposited at a High Se/Bi Flow Rate"

_nanomaterials, 2023, doi:10.3390/nano13202785_

Round 1

Reviewer 1 Report

This paper is an interesting but unstructured paper. It must be thoroughly revised before publication. The following points need serious revision.

1-Title: The title should be "Structural, optical, electrical and thermoelectric properties of Bi2Se3 films deposited at a high Se/Bi flow rate".

2-Abstract :The following sentences do not belong in the abstract but rather in the introduction: "Due to increasing energy demand, low-cost and green energy generation technologies have been a focus of modern-day research. Among several energy-saving and environmentally friendly methods, semiconductor thermoelectric power generation has gained tremendous attention due to its broad application prospects. Traditionally, toxic elements, such as Te and Pb, were used as semiconductor thermoelectric materials. In addition, high-temperature growth processes 12 were required for their synthesis."

3-Experimental: Correct Al2O3(0001). No information is given on the thickness of the films. However, it is important that the thicknesses of the films are the same if their properties are to be compared. What are the H impurities that the thermal post-treatment is designed to remove? Finally, the setup used to measure the optical properties is not described.

4-Results and Discussion:

For better comparability, the energy axis of Figures 3(a1) and 3(a2) should be identical. Explain in the text the origin of the shoulders of the peaks observed in Figures 3(a1) and 3(b1).

The determination of grain size using Scherrer's formula (lines 166 to 176) should be reported in the section "3.1. XRD and Raman Analysis".

Regarding the electrical properties, isn't it paradoxical that the F2 sample with the best crystallinity (Figures 1-4) has the worst conductivity (Figure 7b)? Unless, of course, the electrical measurements made at the surface don't necessarily represent the bulk properties of the material.

Reviewer 2 Report

Bi2Se3 has emerged as a promising thermoelectric material for low-temperature power application (< 600 K). Among all the synthesis methods, plasma-enhanced chemical vapor deposition (PECVD) method has demonstrated great advantages in the large-scale production of 2D materials with a controlled thickness. The main challenges in synthetizing Bi2Se3 are the need for high temperatures and crystalline substrates, which restrict the scalability and compatibility of Bi2Se3 materials with existing manufacturing processes, and the presence of strong n-type doping due to the presence of defects (mainly Se vacancies). Low-temperature growth methods coupled with innovative approach to reduce the defect concentration and increase the film resistivity of Bi2Se3 have the potential to overcome these challenges and enable the integration of Bi2Se3 materials into a wide range of devices and applications. This work describes a high-yield and low-temperature synthesis method for preparing high-quality Bi2Se3 films. Photoelectric and thermoelectric properties, structure, and morphology of the prepared films prepared in a rich Se environment have been studied. In addition, the effect of Se vapor on film growth is evaluated, and results revealed that higher pressure yielded high-performance Bi2Se3-based materials. Thus, the as-synthesized Bi2Se3 films demonstrated high application potential in environmentally friendly large-area thermoelectric devices. Overall the work is quite useful, well-done and it is of interest  for scientific community working in the field. However, few points of this work lacks sufficient explanation, the following should be revised.

To make the work easier to read, some sentences should be rewritten, for instance,

Line 94 -95

should be rewrite as

Ar was used to maintain a positive pressure gas environment in the chamber to prevent the oxidation of the film during the temperature measurement.

line 99

should be rewrite as

Thus, the voltage difference (ΔV) was measured.

Line 107 the authors claims the preferential orientation of films growth, perpendicular to the substrate surface. If the authors could explain a little more what cause of these preferential orientation.

The optical properties, paragraph 3.4, can be shortened for clarity. Figures 5 b1 and 5 b2 can be regrouped

Line 230 , the authors claimed sample F2 has greater mobility than F1 due to its excellent crystallinity and fewer grain boundaries, however morphological characterization, fig 4 AFM images do not support this assumption. Please comment this.

The english quality should be improuved.....

Reviewer 3 Report

Good work. Experimentally well designed. The results are convincing. Figures are clear. I recommend it to publish.

Author Response

Thank you for your recognition of my article.

Round 2

Reviewer 1 Report

The crystalline orientation of the Al2O3 substrate is (001) according to the abstract and not (0001) as stated in the text.

It's still not clear what the authors mean by H impurities. Are they defects in the form of iterstial hydrogen or hydrogen bonds in the crystal?

Finally, the authors need to better explain the competition between charge carrier mobility and concentration, its correlation with structural properties and its consequence on sample conductivity.
